# Optimizing Tumor Detection in Brain MRI with One-Class SVM and Convolutional Neural Network-Based Feature Extraction

**DOI:** 10.3390/jimaging11070207

**Published:** 2025-06-21

**Authors:** Azeddine Mjahad, Alfredo Rosado-Muñoz

**Affiliations:** GDDP, Department Electronic Engineering, School of Engineering, University of Valencia, 46100 Burjassot, Valencia, Spain

**Keywords:** brain tumor detection, CNNs, OCSVM, dimensionality reduction, frequency analysis, feature-based methods, medical imaging

## Abstract

The early detection of brain tumors is critical for improving clinical outcomes and patient survival. However, medical imaging datasets frequently exhibit class imbalance, posing significant challenges for traditional classification algorithms that rely on balanced data distributions. To address this issue, this study employs a One-Class Support Vector Machine (OCSVM) trained exclusively on features extracted from healthy brain MRI images, using both deep learning architectures—such as DenseNet121, VGG16, MobileNetV2, InceptionV3, and ResNet50—and classical feature extraction techniques. Experimental results demonstrate that combining Convolutional Neural Network (CNN)-based feature extraction with OCSVM significantly improves anomaly detection performance compared with simpler handcrafted approaches. DenseNet121 achieved an accuracy of 94.83%, a precision of 99.23%, and a sensitivity of 89.97%, while VGG16 reached an accuracy of 95.33%, a precision of 98.87%, and a sensitivity of 91.32%. MobileNetV2 showed a competitive trade-off between accuracy (92.83%) and computational efficiency, making it suitable for resource-constrained environments. Additionally, the pure CNN model—trained directly for classification without OCSVM—outperformed hybrid methods with an accuracy of 97.83%, highlighting the effectiveness of deep convolutional networks in directly learning discriminative features from MRI data. This approach enables reliable detection of brain tumor anomalies without requiring labeled pathological data, offering a promising solution for clinical contexts where abnormal samples are scarce. Future research will focus on reducing inference time, expanding and diversifying training datasets, and incorporating explainability tools to support clinical integration and trust in AI-based diagnostics.

## 1. Introduction

The early and accurate detection of brain tumors plays a critical role in improving clinical outcomes and enhancing patients’ quality of life. Brain tumors rank among the leading causes of morbidity and mortality worldwide, with approximately 300,000 new cases reported annually [1]. Magnetic Resonance Imaging (MRI) serves as a fundamental diagnostic tool due to its superior spatial resolution and exceptional soft tissue contrast, enabling the detailed visualization of brain abnormalities. Despite these advantages, the accurate interpretation of MRI scans remains challenging due to inter-observer variability and the heterogeneous appearance of tumors.

The brain, one of the most complex organs in the human body, can develop tumors characterized by uncontrolled proliferation of abnormal tissue. According to the American Society of Clinical Oncology, 85% to 90% of brain tumors are classified as central nervous system tumors [2]. Although brain tumors are less prevalent compared with other cancer types, their high mortality rates underscore the importance of early detection to improve treatment efficacy and survival. Furthermore, the incidence of brain tumors has been increasing globally across all age groups in recent years [3].

In clinical diagnosis, medical imaging modalities such as MRI, Computed Tomography (CT), and Positron Emission Tomography (PET) are indispensable for tumor localization and disease progression monitoring. Among these, MRI is the most frequently used modality in clinical practice due to its excellent image quality and enhanced contrast, which facilitate the detection of subtle brain tissue abnormalities [4]. Additionally, MRI’s non-invasive nature and rapid acquisition time make it the preferred choice for routine diagnostic procedures.

Historically, brain tumor detection has relied on conventional image processing techniques such as segmentation algorithms, smoothing filters, and Fourier transforms, which have often required substantial manual intervention. These traditional approaches were limited by factors such as human error, low processing speed, and difficulty in managing the complex patterns inherent in medical imaging data. Moreover, they frequently failed to delineate tumor boundaries accurately, leading to incomplete or imprecise diagnoses. Consequently, the field has witnessed a progressive shift toward semi-automatic and fully automatic methods that offer improved efficiency, consistency, and diagnostic accuracy [5].

Despite advancements in imaging modalities, several challenges persist in brain tumor detection. One of the most pressing issues is data imbalance, as tumor images are significantly less prevalent than healthy brain images, posing challenges for model training and reliable evaluation. Human error remains a concern, especially when radiologists must analyze large volumes of complex imaging data. In addition, traditional techniques often depend on manually engineered features, which introduce subjectivity and limit the capacity to detect subtle or non-obvious patterns [6].

## 2. Related Work

### 2.1. Deep Learning for Tumor Detection

Deep learning, particularly CNNs, has demonstrated remarkable performance in medical image analysis, especially in brain tumor classification. Specialized architectures such as MobileNet, three-dimensional CNNs (3D-CNNs), Parallel Deep CNNs, and Recurrent CNNs have been developed to tackle challenges like tumor morphological variability and computational constraints [1,7,8,9,10]. These deep learning models consistently outperform traditional methods in terms of accuracy, sensitivity, and specificity. However, a major limitation of deep learning approaches is their computational complexity, often involving millions of parameters, which increases the inference time and demands substantial processing resources. Furthermore, they require large, well-annotated, multi-class datasets that are often scarce in clinical settings [11,12].

While numerous studies have proposed both traditional and deep learning-based approaches for brain tumor detection, several critical limitations remain unaddressed. For instance, classical radiomic techniques often rely on handcrafted features and prior assumptions, which may not generalize well across different patient populations or imaging conditions [13]. Additionally, many CNN-based models exhibit a strong performance on small or curated datasets but suffer from overfitting and lack robustness when deployed on heterogeneous or noisy clinical data [14,15]. Furthermore, few studies explore the effectiveness of combining deep features with classical statistical descriptors within a one-class learning framework. The limited generalization capacity and computational overhead of complex CNN architectures also restrict their clinical applicability [16].

Motivated by these gaps, our study investigates a hybrid strategy that incorporates both traditional and deep feature extraction pipelines and evaluates their effectiveness in an OCC setting using OCSVM.

### 2.2. OCSVM for Tumor Detection

To overcome the limitations posed by imbalanced data, unsupervised learning approaches such as One-Class Classification (OCC) have gained considerable attention. Among these, the OCSVM is a widely used technique trained exclusively on normal samples, enabling the detection of deviations as anomalies without requiring labeled pathological data [17]. This property makes OCSVM particularly suitable for applications like brain tumor detection, where abnormal samples are scarce or difficult to obtain.

Significant progress has been made in enhancing one-class learning models by improving feature representation through deep neural networks such as Convolutional Autoencoders (CAEs) and Generative Adversarial Networks (GANs). These models extract robust features that improve the detection of subtle anomalies. OCSVM remains a highly effective and computationally efficient tool in scenarios with limited positive samples, also offering reduced annotation costs due to its minimal supervision requirements [18].

Extensive research has validated the versatility and robustness of OCSVM across various medical anomaly detection tasks, including the identification of nosocomial infections, mammogram abnormalities, and other imbalanced medical datasets. Beyond healthcare, its application extends to cybersecurity and industrial monitoring, underscoring its broad potential for reliable anomaly detection across diverse fields [19].

### 2.3. Proposed Work

This study proposes and evaluates a novel approach for brain tumor detection in MRI images, with two main objectives. The first objective is to explore and compare various feature extraction techniques, including traditional methods such as Dimensionality Reduction and Representation Methods (DR), Transformations and Frequency Analysis (TFA), Feature-Based Methods (FBM), and statistical descriptors, alongside deep learning-based methods using a custom CNN and widely adopted pretrained architectures such as DenseNet, InceptionV3, MobileNet, and VGG16.

The second objective is to employ an unsupervised anomaly detection model based on OCSVM, trained exclusively on healthy brain images, to identify anomalies without requiring labeled pathological data.

The study aims to determine the most effective feature representation to improve anomaly detection performance within this OCC framework.

To fulfill the objectives of this study, the paper is organized as follows: Section 3 outlines the dataset, preprocessing steps, and detailed methodology for feature extraction and classification. Section 4 introduces the CNN and OCSVM algorithms, highlighting their theoretical foundations and relevance in detection tasks. Section 5, Section 6 and Section 7 present the training and evaluation procedures, followed by a discussion of the results and the final conclusions.

## 3. Materials and Methodology

In this study, we propose an approach based on the OCSVM algorithm for brain tumor detection in medical images. This method is particularly useful in scenarios where data from one class (normal or anomalous) are much more abundant or reliable than data from the other class, which is common in brain tumor detection. The following steps outline the preparation and training process of the model:

Training Process:

The training process is broken down into the following steps, as illustrated in Figure 1:

Training Process:Dataset Preparation:
Preprocessed images were divided into two categories: Healthy and Brain tumor.Image Processing and ROI Extraction:
Prior to feature extraction, images underwent preprocessing steps, including normalization and noise reduction.Subsequently, brain region localization and masking techniques were applied to isolate the region of interest (ROI). This ensures that feature extraction and classification focus on relevant anatomical areas, improving model performance and interpretability.Feature Extraction:
Both handcrafted and deep learning-based techniques were employed to extract robust features, capturing texture, shape, and spatial patterns relevant to brain pathology.Model Training:
The OCSVM model was trained to learn the distribution of the healthy class and detect outliers.In parallel, the CNN model was trained in a supervised manner to classify input images into healthy or tumor categories.

### 3.1. Dataset

In this study, the material used for training and evaluating the proposed brain tumor classification system consists of MRI scans sourced from the publicly available Brain Tumor Detection Dataset [20]. This dataset plays a crucial role as the primary input material for our deep learning models and is commonly used in the medical imaging research community.

The dataset includes a total of 3000 labeled MRI images, evenly split into two diagnostic categories (Figure 2):1500 Healthy brain images (no signs of tumor).1500 Brain Tumor images (showing tumors of various types and sizes).

These images represent the essential material for training models to detect and differentiate between healthy and tumorous brain tissue. Since the manual analysis of such complex medical images can be error-prone, the dataset supports the use of automated techniques for improved accuracy and reproducibility.

Before feeding the images into the classification pipeline, a consistent preprocessing routine was applied to enhance model performance and maintain uniformity:Normalization of pixel values to the [0, 1] range to standardize intensity distribution.Resizing each image to a uniform resolution of 150 × 150 × 3, suitable for input into CNNs.Color format conversion to RGB, ensuring compatibility across all model architectures used in the study.

### 3.2. Brain Region Localization and Masking

In the preprocessing phase, the first step involves identifying the region of interest (ROI) within the brain MRI images to focus subsequent analysis on relevant anatomical areas. This step helps exclude irrelevant regions, improving the accuracy and efficiency of the classification process.

To locate the ROI, we applied several image processing techniques. First, image enhancement was performed by adjusting contrast and brightness and applying smoothing filters to reduce noise, as described in [21]. Then, Otsu’s thresholding method was used to binarize the image, effectively separating the brain tissue from the background. Following this, contour detection was carried out [22], identifying the boundaries of significant anatomical structures.

Based on the detected contours, bounding boxes were generated to localize the ROI within the image. Subsequently, a mask was applied to the original brain MRI scan to isolate the targeted region by covering non-essential areas. This process reduces background noise and focuses the classification algorithms on the relevant brain region, such as the prefrontal cortex or hippocampus.

Figure 3 illustrates these steps: the original brain scan, the ROI highlighted by a bounding box, and the final masked ROI extracted for further analysis.

This targeted approach not only improves computational efficiency but also enhances the effectiveness of the downstream classification models by restricting their focus to meaningful image regions, as discussed in [23].

### 3.3. Experimental Design and Feature Extraction Techniques

After applying localization and masking techniques, the resulting image is used as input for various methods aimed at parameter extraction. The extracted features are then used as input for the unsupervised OCSVM classifier. Figure 4 summarizes this workflow, from the image processing stage to feature extraction and classification. It is important to note that, in the case of CNNs, the same model is used both for feature extraction and as a classifier.

Direct Image Classification Using OCSVM:Raw MRI images are fed directly into an OCSVM without feature extraction, establishing a baseline performance.CNN-Based Classification:A CNN model is trained end-to-end to classify images, serving as a deep learning benchmark.Feature Extraction and OCSVM Evaluation:Feature extraction is a critical component for the performance of the OCSVM model in detecting anomalies in medical images. Instead of directly inputting raw images into the model, advanced feature extraction methods are employed to identify and capture the most relevant aspects of the data. This approach enhances the model’s ability to recognize patterns and detect significant deviations, thereby optimizing both the classification accuracy and computational efficiency.The feature extraction techniques utilized in this study are based on state-of-the-art deep learning architectures that capture hierarchical representations of the data. These representations are essential for handling the inherent complexities of medical images, where subtle features are crucial for identifying anomalies, such as brain tumors. The methods used in this study include the following:
CNNs: These networks are pivotal for extracting complex spatial features from images. Their capacity to learn hierarchical representations enables the identification of high-level patterns, making them a vital tool for medical image classification.DR: Techniques like Principal Component Analysis [24] and Embedded [25] are employed to reduce the number of variables in the images while retaining the most relevant features. This helps simplify the classification process and enhances computational efficiency, especially with large datasets.TFA: Techniques, such as Fast Fourier Transform [26] or Wavelet Transform [27] are used to capture features in the frequency domain. These transformations help identify spatial and frequency patterns indicative of anomalies in the images, such as the presence of brain tumors.FBM: This approach focuses on extracting specific features from the images, such as shapes, edges, or textures, that are relevant for detecting abnormalities. These features are then used to feed the OCSVM model, enhancing classification accuracy.

Each of these techniques is employed to extract different aspects of medical images, enabling the OCSVM model to detect anomalies more accurately. The extracted features from the images are then standardized using the StandardScaler (SS) to ensure a mean of zero and a standard deviation of one. This normalization step enhances the performance of the OCSVM classifier by improving the consistency of the extracted features, which is crucial for effective anomaly detection. Mathematically, standardization is defined as follows:$$z_{i}=\frac{x_{i}-\mu }{\sigma }$$
where

μ is the mean of the feature values.σ is the standard deviation.$x_{i}$ is the original feature value.$z_{i}$ is the standardized feature value.

The standardized features are then fed into the classification model to identify potential anomalies or brain tumors. The following section details the specific algorithms implemented for feature extraction and how they contribute to the detection process. All the extracted features are standardized using SS to normalize the data before classification. The OCSVM hyperparameter ν was tuned with values (0.01, 0.1, 0.9) to optimize the sensitivity and specificity, retaining the best configuration per experiment.

#### 3.3.1. Feature Extraction Using CNNs

The CNN architectures selected for feature extraction include both widely recognized pretrained models—VGG16 [28], DenseNet121 [29], InceptionV3 [30], and MobileNetV2 [31] and a custom CNN specifically designed for this study (see Table 1).

For each architecture, feature vectors are obtained from the final activations after a Global Average Pooling (GAP) operation, followed by additional dense layers. This process produces low-dimensional but highly informative feature vectors, which are subsequently used as input for an OCSVM classifier aimed at anomaly detection.

#### 3.3.2. Feature Extraction with DR

Dimensionality Reduction (DR) techniques are essential for managing high-dimensional image data. By reducing redundancy and focusing on the most informative features, these methods improve computational efficiency and enhance model performance. In this study, two main approaches were employed:Principal Component Analysis (PCA): A linear technique that identifies directions (principal components) capturing the highest variance in the data. PCA reduces input space complexity while preserving essential structural information. In our implementation, the number of components was set to 50.Embedding Techniques: Non-linear methods, such as autoencoders, are capable of learning compact, lower-dimensional representations from high-dimensional data while maintaining relevant semantic and structural properties. In this work, t-SNE was employed to project the data into a one-dimensional space for latent structure visualization.

#### 3.3.3. Feature Extraction with TFA

To extract features in the frequency domain of images, three mathematical transformations were applied: Fast Fourier Transform (FFT), Wavelet Transform, and Hartley Transform. For each transformation, a feature vector was computed as follows:FFT: A 2D discrete Fourier transform was applied to grayscale images. The resulting magnitude spectrum was flattened to form a feature vector representing the image’s frequency content.Wavelet Transform: A discrete wavelet transform (DWT) was applied using the Haar wavelet up to decomposition level 1. The approximation coefficients (cA) and the horizontal, vertical, and diagonal detail coefficients (cH, cV, cD) were concatenated and flattened to form the feature vector.Hartley Transform: An approximation of the Hartley Transform was performed using the Discrete Cosine Transform (DCT) with orthonormal normalization. The resulting coefficients were flattened to produce the final feature vector.

#### 3.3.4. Feature Extraction with FBM

To capture basic statistical and structural information from the images, two types of handcrafted features were extracted: histogram-based and gradient-based features.

Histogram Features: The intensity histogram was computed for each color channel (R, G, B) in images normalized to the [0, 255] range. Each histogram was normalized and flattened to form a feature vector. The number of bins used was set to NUM_BINS.Gradient Features: The Sobel operator was applied to grayscale images to compute horizontal and vertical gradients. The gradient magnitude was calculated as the square root of the sum of squared gradients in both directions, and then flattened to generate the feature vector.

After applying the different techniques, Table 2 summarizes the evaluated methods and algorithms, including input image dimensions and the number of features extracted per image, i.e., the output size resulting from each feature extraction technique.

## 4. Machine Learning: Supervised and Unsupervised Approaches

Machine learning encompasses a wide range of techniques that enable systems to automatically learn from data and make predictions or decisions without being explicitly programmed. In this study, we explore both supervised and unsupervised approaches tailored to the task of brain tumor detection using MRI data. Specifically, we compare the performance of a custom-designed CNN with that of the OCSVM algorithm.

### 4.1. Supervised Learning: CNNs

CNNs have emerged as powerful models for analyzing visual data, especially in biomedical image classification. Their layered architecture enables automatic extraction of spatial features critical for recognizing tumor structures. Figure 5 shows a simplified view of our proposed CNN pipeline [23].

Input Layer: Processes pre-processed MRI slices as input tensors.Convolutional Blocks: Composed of filters (kernel sizes 3×3 and 5×5) to capture spatial features, such as tumor boundaries and tissue patterns.Non-linear Activation: ReLU function introduces non-linearity, enabling the model to approximate complex mappings.Downsampling Layers: Max pooling (2×2) layers reduce dimensionality while preserving key spatial features.Regularization Units: Dropout layers randomly deactivate a percentage (30–50%) of neurons during training to mitigate overfitting.Dense Layers: Fully connected layers integrate learned features for global representation and decision making.Classification Layer: A final dense layer with Softmax (for multi-class) or Sigmoid (for binary) activation outputs the classification score.

#### Hyperparameter Tuning

Model performance is heavily dependent on careful selection of hyperparameters, including network depth, number of filters, activation functions, learning rate, batch size, and dropout rate. These parameters were optimized experimentally using validation accuracy as the guiding metric.

### 4.2. Unsupervised Learning: OCSVM

OCSVM is an unsupervised anomaly detection algorithm that constructs a decision boundary enclosing the majority of data points, effectively identifying outliers (i.e., abnormal tumor cases). Unlike CNNs, OCSVM does not require labeled data, making it suitable for medical contexts with limited annotations [19].

#### 4.2.1. Mathematical Formulation

Given a training dataset ${\left\{x_{i}\right\}}_{i=1}^{n}$ where $x_{i}\in R^{d}$, OCSVM aims to find a function f(x) such that most data points lie inside the decision region. The optimization objective is as follows: (1)$$\min_{w,\rho ,\xi }\frac{1}{2}{∥w∥}^{2}+\frac{1}{\nu n}\sum_{i=1}^{n}\xi_{i}-\rho$$
subject to the following:(2)$$w\cdot \phi \left(x_{i}\right)\geq \rho -\xi_{i},\;\xi_{i}\geq 0,\;i=1,\ldots ,n$$
where
*w* is the normal vector to the separating hyperplane,ϕ(x) is the kernel-induced mapping to a higher-dimensional space,ρ is the offset (threshold),$\xi_{i}$ are slack variables that allow for soft margins,ν∈(0,1] is a user-defined parameter that controls the fraction of outliers and margin.

#### 4.2.2. Kernel Functions

OCSVM leverages the kernel trick to handle non-linear boundaries. Common kernels include the following:Linear: $K\left(x_{i},x_{j}\right)=x_{i}\cdot x_{j}$Radial Basis Function (RBF): $K\left(x_{i},x_{j}\right)=\exp\left(-\frac{∥x_{i}-x_{j}{∥}^{2}}{2\sigma^{2}}\right)$Polynomial: $K\left(x_{i},x_{j}\right)={\left(x_{i}\cdot x_{j}+c\right)}^{d}$

#### 4.2.3. Relevance in Medical Imaging

Extensive research has validated the versatility and robustness of OCSVM in various medical anomaly detection scenarios, including detection of nosocomial infections, mammogram abnormalities, and other imbalanced datasets [19]. Its ability to learn from data with minimal supervision makes it ideal for early-stage diagnostic tools where labeled data is scarce.

## 5. Model Training and Evaluation

This section presents the architecture, experimental setup, and evaluation metrics used for the CNN and OCSVM models applied to brain image analysis.

### 5.1. CNN-Based Supervised Classification

#### 5.1.1. Model Architecture

The proposed CNN model consists of nine layers, including three convolutional layers, three max-pooling layers, a flattening layer, and three fully connected (dense) layers. The convolutional layers use progressively increasing numbers of filters (32, 64, and 128) with kernel sizes of 3 × 3. Each convolutional layer is followed by a max-pooling operation to reduce the spatial dimensions while preserving the most relevant features. After flattening the multidimensional output, it is passed through dense layers to perform the final classification. The final output layer uses a sigmoid activation function suitable for binary classification. The detailed architecture of the CNN model is summarized in the subsubsection “CNN Model Architecture” (Section 3.3.1), as shown in Table 1.

#### 5.1.2. Experimental Setup

In the supervised classification experiments, the CNN model was trained to distinguish between healthy and tumorous brain images. The dataset was divided into training and testing subsets using a balanced class distribution:Training set: 80% of the healthy images and 80% of the tumorous images.Test set: 20% of the healthy images and 20% of the tumorous images.Number of epochs: 100

This configuration ensures that both training and test sets are balanced across classes, allowing the model to learn representative features from both categories. The model was trained using the binary cross-entropy loss function and optimized with the Adam optimizer.

The model’s performance was evaluated using several key metrics such as precision, recall, F1-score, specificity, and accuracy, as described in Section 5.3. These metrics provide a comprehensive understanding of how well the model distinguishes between healthy and tumorous cases.

The evolution of training and validation accuracy and loss across epochs is shown in Figure 6, providing insights into the convergence behavior and generalization performance of the CNN model. The left plot demonstrates the progression of training and validation accuracy over the 100 epochs, while the right plot shows the corresponding loss values, indicating how the model learned to minimize the loss during training.

### 5.2. OCSVM for Anomaly Detection

The dataset was divided as follows:Training: 80% of the features extracted from healthy images used in the CNN training.Test: Remaining healthy features and all features from abnormal images, none of which were used during OCSVM training.

This setup simulates a realistic anomaly detection scenario where the OCSVM model is trained exclusively on healthy data. Five-fold cross-validation was performed to assess the model’s robustness, and the reported results correspond to the average performance across these runs.

### 5.3. Performance Metrics

The performance of both the CNN and OCSVM models was evaluated using their respective test datasets. The evaluation focused on the following metrics: Accuracy (Acc), Sensitivity (Sens), Precision (Pr), F1-score (F1), and the Area Under the ROC Curve (AUC). The formulas used to compute each metric are as follows:(3)$$\mathrm{Accuracy}\left(\%\right)=\frac{TP+TN}{TP+TN+FP+FN}\times 100$$(4)$$Sensitivity\left(\%\right)=\frac{TP}{TP+FN}\times 100$$(5)$$\mathrm{Precision}=\frac{TP}{TP+FP}\times 100$$(6)$$F1=2\times \frac{\mathrm{Precision}\times \mathrm{Recall}}{\mathrm{Precision}+\mathrm{Recall}}\times 100$$(7)$$\mathrm{TPR}=\frac{TP}{TP+FN},\;\mathrm{FPR}=\frac{FP}{FP+TN}$$(8)$$\mathrm{AUC}=\int_{0}^{1}\mathrm{TPR}\left(\mathrm{FPR}\right)\;d\left(\mathrm{FPR}\right)$$
where TP (true positives), TN (true negatives), FP (false positives), and FN (false negatives) are the classification outcomes. The AUC (Area Under the Curve) corresponds to the area under the Receiver Operating Characteristic (ROC) curve, which plots the True Positive Rate (TPR) against the False Positive Rate (FPR). A higher AUC indicates better model performance in distinguishing between classes.

## 6. Results and Discussion

This section presents and analyzes the results obtained from the different approaches implemented for image detection and classification. We compare methods based on the direct use of images, feature extraction through CNNs, and traditional signal processing techniques, evaluating their performance using relevant metrics. Each subsection details specific experiments and is accompanied by tables summarizing the quantitative outcomes.

The tables include standard classification metrics such as precision, sensitivity, F1 score, and accuracy, along with the absolute counts of true positives, false negatives, false positives, and true negatives for each class. The global accuracy is identical for both classes because it reflects the total proportion of correct classifications across all samples.

### 6.1. Direct Image-Based Models with OCSVM and CNN

In this section, we present and analyze the performance of different classification models applied directly to raw medical images. The aim is to evaluate the effectiveness of using standard pixel-level input, without handcrafted or deep feature extraction, for distinguishing between healthy and tumorous samples. The results provide insight into the strengths and limitations of traditional classifiers like OCSVM when compared with data-driven models such as CNN.

As shown in Table 3, OCSVM applied directly to raw images yielded poor performance, particularly in detecting tumorous cases. With a sensitivity of only 47.34% for the tumorous class and a global accuracy of 63.04%, the model struggled to generalize from pixel-level intensities. These results suggest that OCSVM is not well suited to this type of high-dimensional input without prior feature learning or reduction.

In contrast, the CNN model demonstrated substantially better performance. It achieved an accuracy of 97.83%, with high and balanced precision and recall values across both classes. This confirms the capacity of CNNs to learn complex visual patterns from raw data, making them especially suitable for medical image analysis tasks where spatial features are critical.

To visualize the classification behavior of the CNN model, Figure 7 displays the confusion matrix obtained on the test set. The model correctly classified 396 healthy and 279 tumorous cases, with only 15 misclassifications out of 690 total instances. This strong performance—characterized by low false positive and false negative rates—demonstrates the model’s robustness and reliability, which are crucial attributes for clinical diagnostic systems.

These initial results serve as a baseline for evaluating the performance of more advanced architectures and hybrid techniques discussed in the following sections.

### 6.2. Feature Extraction Using Deep Learning + OCSVM

The performance metrics for the classification of healthy and tumorous cases using various deep learning algorithms are summarized in Table 4.

All the models achieved high precision and sensitivity values for the healthy class, with DenseNet121 and VGG16 showing the best overall performance, reaching precision values above 91% and sensitivities exceeding 99%. Tumorous class detection was more variable; DenseNet121 and VGG16 also led with precision above 98% and sensitivity close to 90%.

MobileNetV2 demonstrated competitive results with precision and sensitivity above 86% for the tumorous class. The CNN model, used as a baseline, showed lower performance compared with the deep feature extraction models, particularly in the tumorous class sensitivity (68.29%), indicating a higher rate of false negatives.

Overall accuracy values were consistent with the trends in precision and sensitivity, with VGG16 and DenseNet121 achieving the highest accuracies above 94%.

The confusion matrix components (TP, FN, FP, TN) further illustrate the models’ effectiveness in distinguishing between healthy and tumorous samples, confirming the superior discriminative capacity of models like DenseNet121 and VGG16.

### 6.3. Benchmarking Traditional Feature Extraction with OCSVM

To compare the proposed approach, several classical feature extraction methods combined with OCSVM were evaluated. Table 5, Table 6 and Table 7 show the performance metrics obtained for each method, reporting values for both Healthy and Tumorous classes.

Table 5 presents the performance of two dimensionality reduction methods—Embedded and PCA combined with OCSVM—for classifying healthy and tumorous cases.

Embedded + OCSVM achieved high sensitivity (98.61%) and moderate precision (21.82%) for the tumorous class, indicating that the model is highly sensitive to anomalies but produces many false positives. For the healthy class, the model demonstrated excellent precision (99.18%) but very low sensitivity (32.40%), meaning that while most predicted healthy cases are correct, it fails to identify many true healthy cases. The F1-scores further reflect this imbalance (48.91% for healthy, 35.74% for tumorous). The overall accuracy was low (43.04%), indicating poorly balanced classification across both classes. This configuration tends to overclassify normal cases as anomalies, resulting in many false alarms that reduce its clinical reliability.

PCA + OCSVM, on the other hand, presented moderate precision (81.17%) and sensitivity (74.46%) for the healthy class, showing a better capacity to detect normal samples. However, its performance on the tumorous class was still very poor, with precision at 6.82% and sensitivity at 9.76%, resulting in a very low F1-score (8.01%) for anomalies. Although the overall accuracy was higher (64.11%) than that of Embedded + OCSVM, this is misleading, as the classifier is clearly biased towards the majority class (healthy).

Overall, these results indicate that neither method achieves balanced performance across both classes. This highlights the limitations of traditional dimensionality reduction approaches combined with one-class classification, and emphasizes the need for more powerful techniques—such as deep learning-based feature extraction—to achieve clinically reliable tumor detection.

Table 6 compares the performance of gradient- and histogram-based feature extraction methods combined with OCSVM.

The Gradient + OCSVM method shows a high sensitivity (86.75%) for the tumorous class, indicating some ability to detect anomalies. However, its precision is very low for both classes—only 13.20% for the healthy class—reflecting many false positives and negatives. The overall accuracy is just 25.00%, close to the random classification, highlighting poor discriminative power.

Histogram + OCSVM achieves better results for the healthy class, with high sensitivity (93.40%) and precision (83.81%), indicating the strong detection of normal samples. Yet, its performance in the tumorous class remains weak, with both sensitivity and precision below 23%, showing difficulty in detecting anomalies. Despite a higher overall accuracy of 79.44%, this is influenced by class imbalance and does not fully represent the model’s effectiveness for the minority class.

In summary, both traditional feature-based methods face challenges in imbalanced and anomaly detection tasks, reinforcing the need for deep learning approaches to improve tumor classification accuracy and robustness.

The evaluation metrics for FFT, Wavelet, and Hartley methods combined with OCSVM (see Table 7) reveal significant differences in performance across these classical frequency domain techniques.

Hartley + OCSVM showed extremely low sensitivity for the healthy class, correctly detecting only 4.00%, which implies a very high number of false negatives. Although its precision for the healthy class was 71.43%, this value is misleading due to the overwhelming imbalance and misclassification. The tumorous class, on the other hand, was recognized with a high sensitivity of 91.63%, suggesting effective detection of anomalies. However, the very low precision of 15.45% again reflects excessive false positives. The overall accuracy reached only 18.07%, indicating poor general performance.

Wavelet + OCSVM showed similar patterns. For the healthy class, it reached a sensitivity of 4.27% and a precision of 64.64%, with a modest F1-score of 7.91%. Meanwhile, the tumorous class had a sensitivity of 87.80% and a precision of 14.93%, yielding an F1-score of 25.46%. Despite this, the overall accuracy remained low at 17.68%, again due to imbalance and poor classification of the majority class.

FFT + OCSVM demonstrated slightly better overall behavior. For the healthy class, sensitivity was 8.27%, with a precision of 90.51% and an F1-score of 15.05%. The tumorous class was recognized with a high sensitivity of 88.50% and a precision of 15.58%, leading to an F1-score of 26.42%. Still, the overall accuracy was just 21.14%, limited by the model’s poor detection of healthy instances.

In summary, while these traditional frequency-domain methods combined with OCSVM show some effectiveness in identifying tumorous cases—especially FFT and Hartley—they all suffer from very low sensitivity for the healthy class and high false positive rates. This imbalance severely limits their practical utility. These findings underscore the advantage of deep learning-based feature extraction techniques, which can offer more balanced, robust, and clinically reliable tumor detection.

### 6.4. Execution Time and Computational Complexity

Table 8 presents a comparison of the average feature extraction time, prediction time per image, and computational complexity (measured in FLOPs) for each deep learning backbone combined with OCSVM.

Among the evaluated models, MobileNetV2 demonstrates the highest efficiency in terms of inference speed, requiring only 4.64 ms for feature extraction and 0.012 ms for prediction per image. This makes MobileNetV2 highly suitable for real-time or resource-constrained applications.

In contrast, VGG16 shows the longest feature extraction time at 91.09 ms, which could hinder its deployment in latency-sensitive scenarios despite its potentially higher accuracy. DenseNet121 and InceptionV3 offer a balance with moderate extraction times and computational demands.

Regarding computational complexity, VGG16 demands the most FLOPs ($1.32\times 10^{10}$), reflecting its deeper and more complex architecture. InceptionV3 and DenseNet121 require a moderate number of FLOPs ($2.52\times 10^{9}$), while MobileNetV2 stands out as the most lightweight model, with just $2.93\times 10^{8}$ FLOPs, confirming its advantage for embedded and mobile applications.

It is important to note that the reported prediction times correspond to the OCSVM classifier when combined with the deep learning backbone features. For models operating as pure CNN classifiers without OCSVM, the prediction time reflects the direct inference time of the CNN. Additional processing steps, such as filtering, localization, or masking, are not included in these timings as they are considered separate from feature extraction and classification.

Overall, these metrics highlight the trade-offs between accuracy, speed, and computational cost, offering valuable guidance for selecting appropriate models based on the constraints and requirements of specific deployment environments.

### 6.5. Visual Evaluation and Additional Metrics

The visual evaluation presented in Figure 8, Figure 9, Figure 10, Figure 11 and Figure 12 complements the quantitative results by showing the ROC curves, true versus predicted label distributions, and prediction accuracy distributions for all evaluated models.

Models with higher AUC values in the ROC curves, such as the CNN (AUC = 0.97), DenseNet121 (AUC = 0.95), and VGG16 (AUC = 0.95), also exhibit strong alignment between true and predicted labels, confirming robust discriminative capability. Their prediction accuracy distributions reveal a high proportion of correct classifications and a very small angle of error, reflecting precise model performance.

In contrast, traditional feature-based methods like PCA (AUC = 0.42), Gradient (AUC = 0.46), Histogram (AUC = 0.52), FFT (AUC = 0.48), Hartley (AUC = 0.46), and Wavelet (AUC = 0.46) demonstrate poor ROC performance, which is further corroborated by low similarity in the true vs predicted plots and wider error angles in the accuracy distributions. These observations indicate limited anomaly detection ability and less reliable classification.

Intermediate results, such as those from MobileNetV2 + OCSVM (AUC = 0.92) and InceptionV3 + OCSVM (AUC = 0.94), suggest a balance between computational efficiency and predictive accuracy, which is consistent across all three visual representations.

Overall, these graphical analyses reinforce the superiority of deep learning backbone networks combined with OCSVM for tumor classification tasks over traditional feature extraction methods, both in terms of accuracy and reliability.

### 6.6. Analysis of Methods and Comparison with Other Studies

To evaluate the effectiveness of the proposed method, a comparative analysis was conducted against existing approaches that use similar datasets.

Table 9 provides a detailed summary of the performance metrics from various studies, allowing for a clear comparison of the accuracy, precision, and other relevant measures. Ismael et al. [32] introduced a neural network framework tailored for brain tumor classification. By incorporating statistical descriptors within the architecture, their model achieved a diagnostic accuracy of 91.9%, showcasing the importance of statistical features for enhancing performance. Cheng et al. [33] proposed a strategy aimed at enhancing brain tumor classification accuracy through the combined use of data augmentation and region partitioning. Their evaluation involved techniques such as intensity histograms, grey-level co-occurrence matrices (GLCM), and a bag-of-words (BoW) method. Applied to a dataset consisting of 3064 brain MRI scans, their approach achieved a classification accuracy of 91.3%. Afshar et al. [34] presented a capsule network architecture (CapsNet) for brain tumor classification. This model improved precision by capturing spatial hierarchies between tumors and surrounding structures, addressing a limitation of conventional CNNs. CapsNet achieved an accuracy of 86.7% with segmentation and 78% without it, outperforming other models, such as those in [35,36,37]. Tahir et al. [35] developed a classification method using MRI data, which employed a two-dimensional discrete wavelet transform (DWT) combined with Daubechies wavelet-based features. This approach demonstrated an accuracy of 86%, emphasizing the significance of wavelet-based spatial information in medical image analysis. Paul et al. [38] leveraged deep learning, specifically CNNs, to create a model for brain tumor classification. Their approach achieved a classification accuracy of 90.3%, and they noted that reducing image resolution could improve training efficiency, benefiting clinical workflows. A refined version of CapsNet was later introduced by Afshar et al. [39], offering superior performance to CNN-based models while requiring fewer training samples. This second-generation CapsNet demonstrated robustness to variations, such as affine transformations and rotations, achieving a classification accuracy of 90.9%. However, its reliance on segmented data added to its architectural complexity. Zhou et al. [40] employed a method that utilized automated region segmentation guided by recurrent neural networks. This technique effectively identified axial slices for classification, achieving an accuracy of 92.1%, which confirmed its suitability for clinical tumor detection. Pashaei et al. [41] adopted a hybrid approach, using CNNs for feature extraction and a Kernel Extreme Learning Machine (KELM) for classification. Their model achieved an impressive accuracy of 93.7%, outperforming other machine learning algorithms, such as SVM, KNN, and RBFNN. Zhou et al. [40] employed a method that utilized automated region segmentation guided by recurrent neural networks. This technique effectively identified axial slices for classification, achieving an accuracy of 92.1%, which confirmed its suitability for clinical tumor detection. In a separate approach, Abiwinanda et al. [36] implemented several CNN architectures, totaling seven, without using image segmentation. Among these, the second configuration outperformed the others, achieving a training accuracy of 98.5% and a testing accuracy of 84.2%. Despite these impressive results, simpler CNN models often struggle to capture complex patterns, limiting overall performance due to their lack of expressive power. Navid et al. [42] proposed a multi-class classification system utilizing a generative adversarial network (GAN), with a neural network functioning as the discriminator. This GAN-based architecture augmented the dataset and helped mitigate overfitting. Modified fully connected layers and tumor classification tasks were trained using 5-fold cross-validation, yielding accuracies of 93.0% for inserted splits and 95.6% for random splits. Guo et al. [37] combined CNNs with graph-based representations, applying the model to PET scans for Alzheimer’s disease classification. Their CNN-graph hybrid achieved an accuracy of 93% in binary classification and 77% in three-class classification on the ADNI dataset, demonstrating its strength in neuroimaging-based diagnostics.

In this study, the pure CNN model achieved an accuracy of 97.83%, while the VGG16 network combined with OCSVM obtained 95.33%. Both results significantly outperform several methods previously reported in the literature. These figures demonstrate the effectiveness of deep neural network-based approaches for brain tumor classification. The CNN stands out for its ability to robustly extract discriminative features directly from magnetic resonance images, whereas the combination of VGG16 with OCSVM provides a hybrid strategy that can enhance detection in specific contexts. Together, these results highlight the great potential of deep learning techniques to improve automated and early diagnosis of brain tumors in clinical settings.

### 6.7. Discussion

The results clearly demonstrate that using deep learning models as feature extractors significantly enhances anomaly detection performance compared with directly using raw images or traditional handcrafted features.

In particular, DenseNet121 and VGG16 provided an excellent balance between accuracy, sensitivity, and computational efficiency. DenseNet121 achieved an accuracy of 98.45% and a sensitivity of 89.90% for detecting anomalies, substantially outperforming traditional approaches like PCA + OCSVM, which only reached approximately 10.0% accuracy and 0.1% recall. This highlights the advantage of deep feature representations in capturing complex data patterns relevant to anomalies.

The choice of the ν parameter in OCSVM had a critical impact on sensitivity. Lower ν values tended to bias the model toward the majority (normal) class, while higher values improved anomaly detection capabilities, as observed with Wavelet + OCSVM (ν=0.9). This illustrates the importance of careful hyperparameter tuning in OCC.

From a computational perspective, MobileNetV2 demonstrated superior efficiency, with a total feature extraction plus prediction time of only 0.00464 s per image. This suggests MobileNetV2 is highly suitable for real-time or resource-constrained environments where speed is critical.

ROC curve analyses reinforced these findings, with models that effectively separated classes achieving higher AUC scores. However, some models exhibited signs of slight overfitting, performing exceptionally well on the normal class but showing reduced generalization to anomalies. This suggests potential room for improvement in regularization or data augmentation strategies.

Traditional handcrafted feature-based methods (Histogram, Gradient, FFT, Hartley) and classical dimensionality reduction techniques like PCA and Embedded feature extraction generally showed poor performance in both detection accuracy and sensitivity. Their inability to capture intricate data structures likely led to low discriminative power, consistent with the relatively poor ROC and prediction-actual label comparisons.

In summary, deep learning-based feature extraction methods consistently outperformed other approaches across all evaluation metrics, underscoring their effectiveness and value for unsupervised anomaly detection tasks in complex image datasets.

### 6.8. Application in Real Clinical Settings

Artificial intelligence (AI), particularly convolutional neural networks (CNNs), holds significant promise for improving patient care by enabling automated tumor detection in medical imaging. By analyzing MRI, CT, and other imaging modalities, AI supports radiologists in diagnostic workflows through accurate and efficient identification of abnormalities. AI models trained on tumor patterns can outperform manual interpretations, providing valuable assistance in clinical decision-making. Moreover, AI-driven monitoring systems can continuously assess imaging data in critical care, promptly alerting healthcare professionals to potential tumors—an especially vital capability in intensive care units.

In addition to enhancing diagnostic accuracy, AI-based detection improves healthcare efficiency by facilitating patient prioritization according to clinical urgency. Successful deployment requires addressing challenges, such as rigorous clinical validation and regulatory approval, to ensure patient safety. Close collaboration among radiologists, data scientists, and regulatory bodies is essential for responsible and safe integration of AI into routine practice.

Recent advances in deep neural networks and dimensionality reduction techniques have shown strong potential for precise tumor localization. Combining feature extraction methods like the Fast FFT and Wavelet Transform with OCC approaches has proven effective for anomaly detection while maintaining low computational demands.

Ongoing research is crucial to refine these algorithms further. This study demonstrates that real-time tumor detection can be achieved without heavy computational overhead, indicating that the approach is adaptable to diverse medical imaging modalities for clinical use.

## 7. Conclusions

This study presents an effective strategy for brain tumor detection based on the integration of deep learning models for feature extraction with OCC through OCSVM. The experimental results, summarized in Table 4, demonstrate that deep architectures, such as DenseNet121 and VGG16, achieve a robust balance between accuracy, precision, and sensitivity. DenseNet121 achieved an overall accuracy of 94.83%, with a precision of 99.23% and a sensitivity of 89.97% for tumor detection—outperforming other models like ResNet50 and CNN with accuracies below 85%. VGG16 also showed competitive performance (accuracy of 95.33%, precision of 98.87%, sensitivity of 91.32%).

Furthermore, MobileNetV2 offered a strong trade-off between efficiency and accuracy (92.83%), with the lowest computational cost, making it well-suited for real-time or resource-constrained applications. Interestingly, the standalone CNN model achieved a high accuracy of 97.83% without OCSVM, underscoring the strong discriminative power of convolutional networks trained end-to-end.

These findings confirm that deep learning-based feature extraction substantially outperforms traditional handcrafted methods in the context of unsupervised brain tumor detection. The hybrid approach of combining CNN-based features with OCSVM proved particularly useful in imbalanced or semi-supervised scenarios.

This work highlights the clinical relevance of AI systems in supporting early and accurate diagnosis of brain tumors, particularly when integrated into medical imaging workflows. However, future studies should focus on improving generalization to unseen data, optimizing inference speed, and validating the model across multicenter, heterogeneous datasets. Moreover, integrating explainability techniques and real-time feedback mechanisms will be critical for increasing physician trust and facilitating regulatory approval.

In conclusion, the results reinforce the potential of deep learning methods not only to enhance diagnostic accuracy but also to pave the way for safer, faster, and more accessible clinical decision-making tools in oncology and beyond. 

## Figures and Tables

**Figure 1 jimaging-11-00207-f001:**
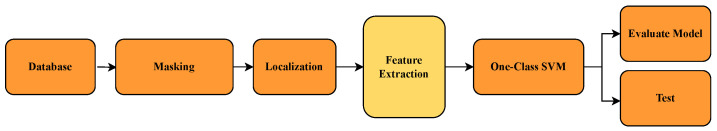
Testing Graph: Model performance during the testing phase.

**Figure 2 jimaging-11-00207-f002:**
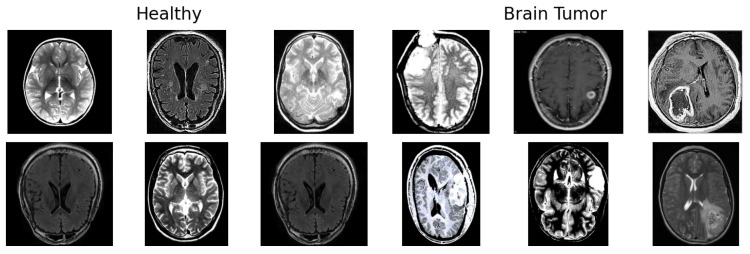
Example images from the two classes in the dataset: Brain Tumor and Healthy.

**Figure 3 jimaging-11-00207-f003:**
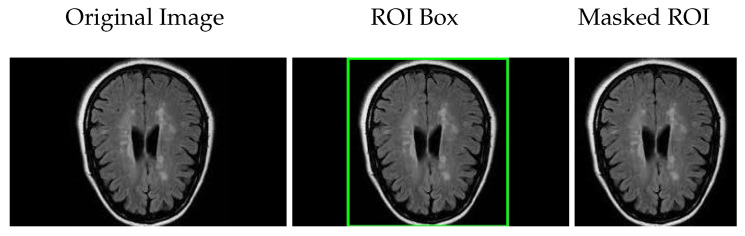
Brain region localization: original brain scan, ROI marked with a bounding box, and the extracted region using a mask.

**Figure 4 jimaging-11-00207-f004:**
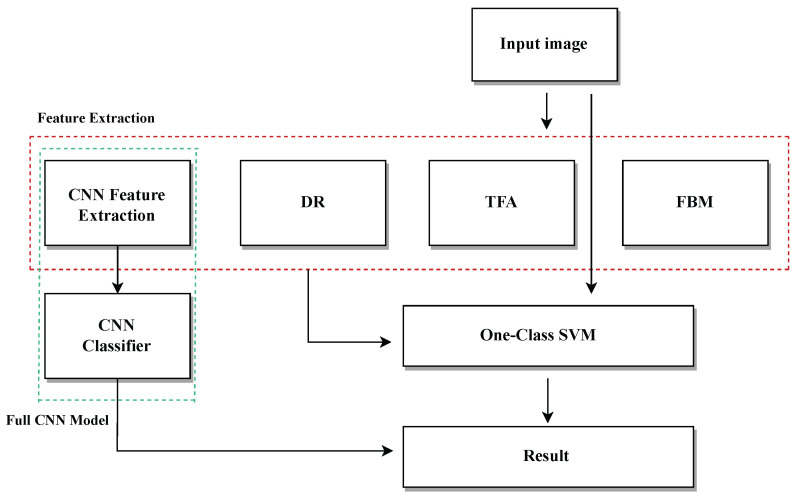
Overview of the evaluation pipeline, including preprocessing, feature extraction, and classification stages. CNNs are used both for feature extraction and classification.

**Figure 5 jimaging-11-00207-f005:**
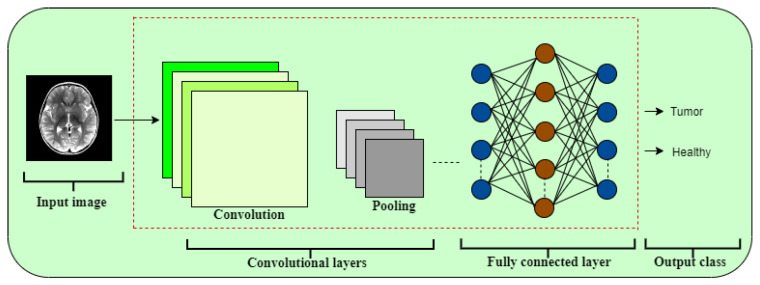
General architecture of the CNN model used for tumor classification.

**Figure 6 jimaging-11-00207-f006:**
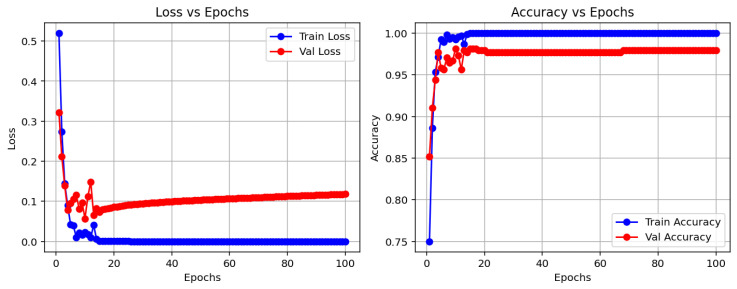
Training and validation accuracy and loss across epochs. The left plot shows the progression of accuracy, while the right plot shows the loss values for both training and validation.

**Figure 7 jimaging-11-00207-f007:**
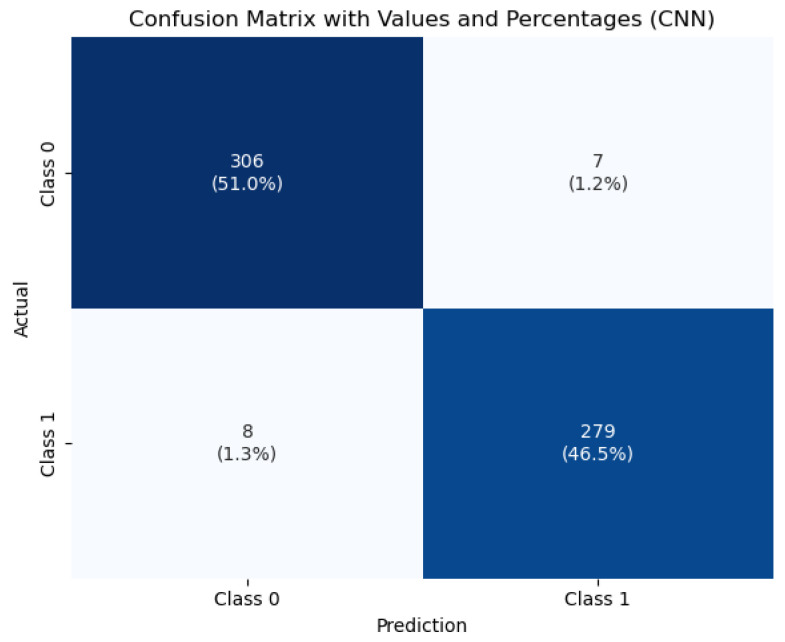
Confusion matrix of the CNN model for tumor classification.

**Figure 8 jimaging-11-00207-f008:**
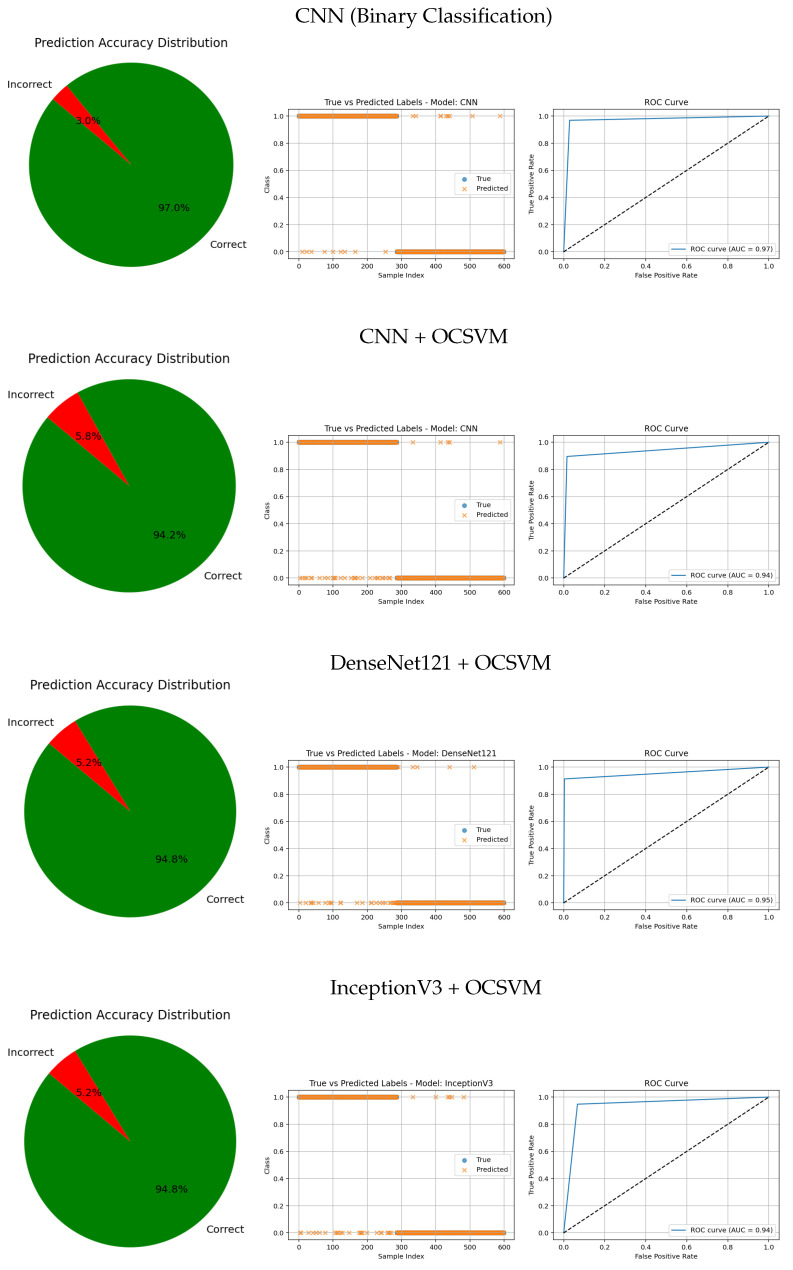
Comparative results of various CNN architectures and CNN-based anomaly detection approaches. Each row shows the prediction accuracy distribution, predicted labels, and ROC curve.

**Figure 9 jimaging-11-00207-f009:**
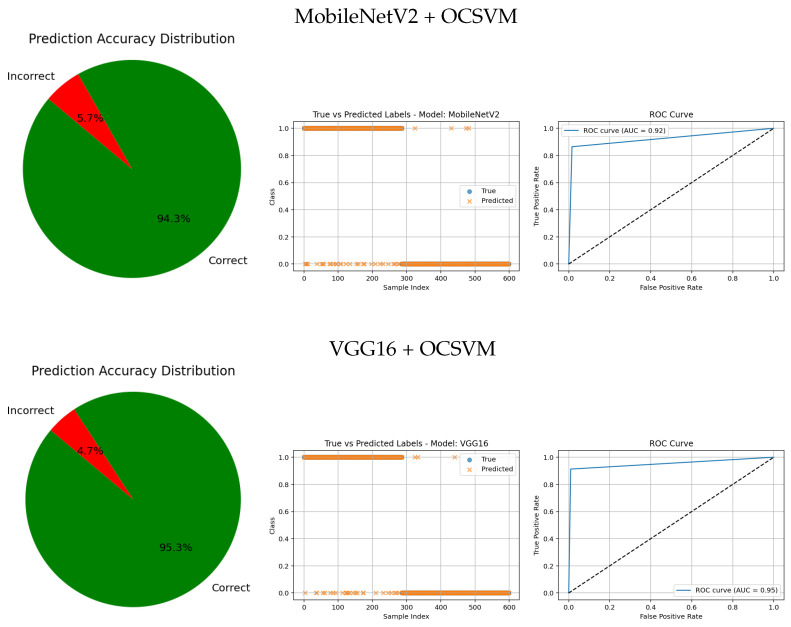
Comparative results of CNN architectures combined with anomaly detection approaches. Each row shows the prediction accuracy distribution, predicted labels, and ROC curve.

**Figure 10 jimaging-11-00207-f010:**
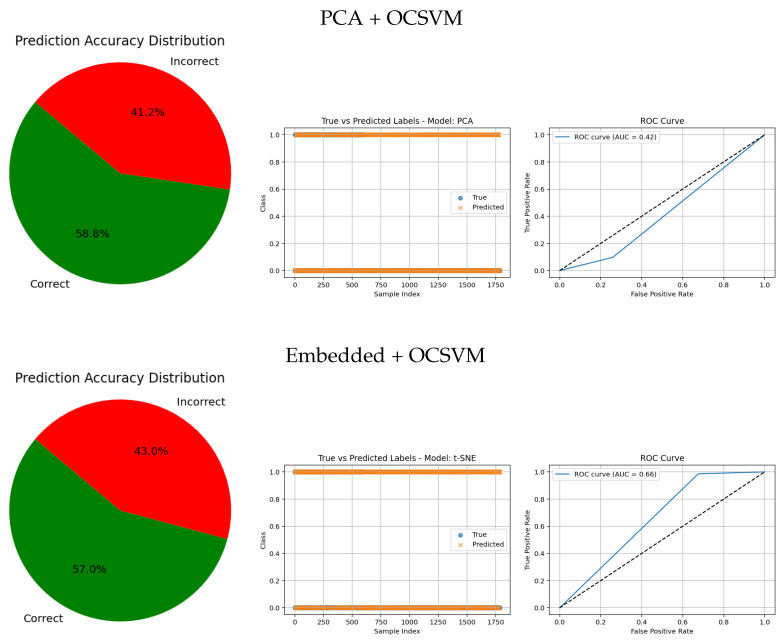
Comparative results of traditional feature-based methods combined with anomaly detection approaches. Each row shows the prediction accuracy distribution, predicted labels, and ROC curve.

**Figure 11 jimaging-11-00207-f011:**
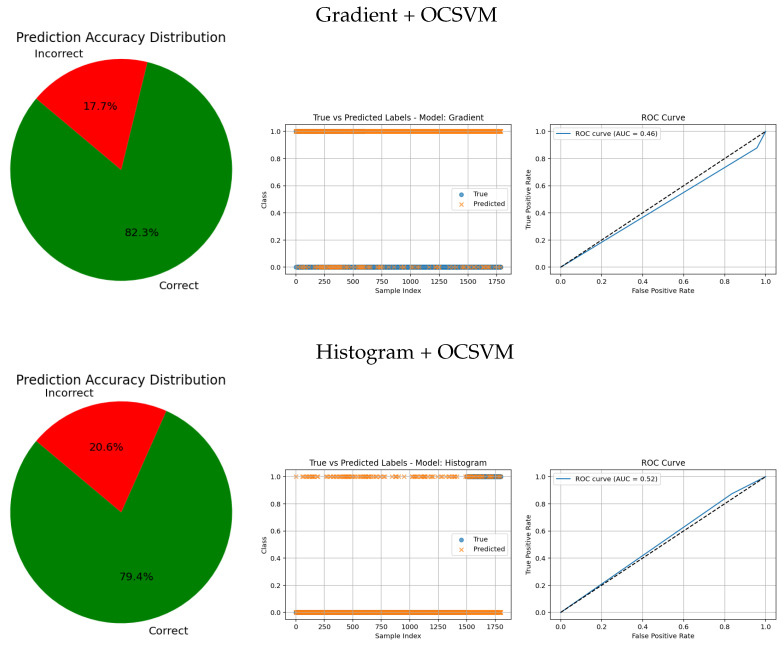
Comparative results of traditional feature-based methods combined with anomaly detection approaches. Each row shows the prediction accuracy distribution, predicted labels, and ROC curve.

**Figure 12 jimaging-11-00207-f012:**
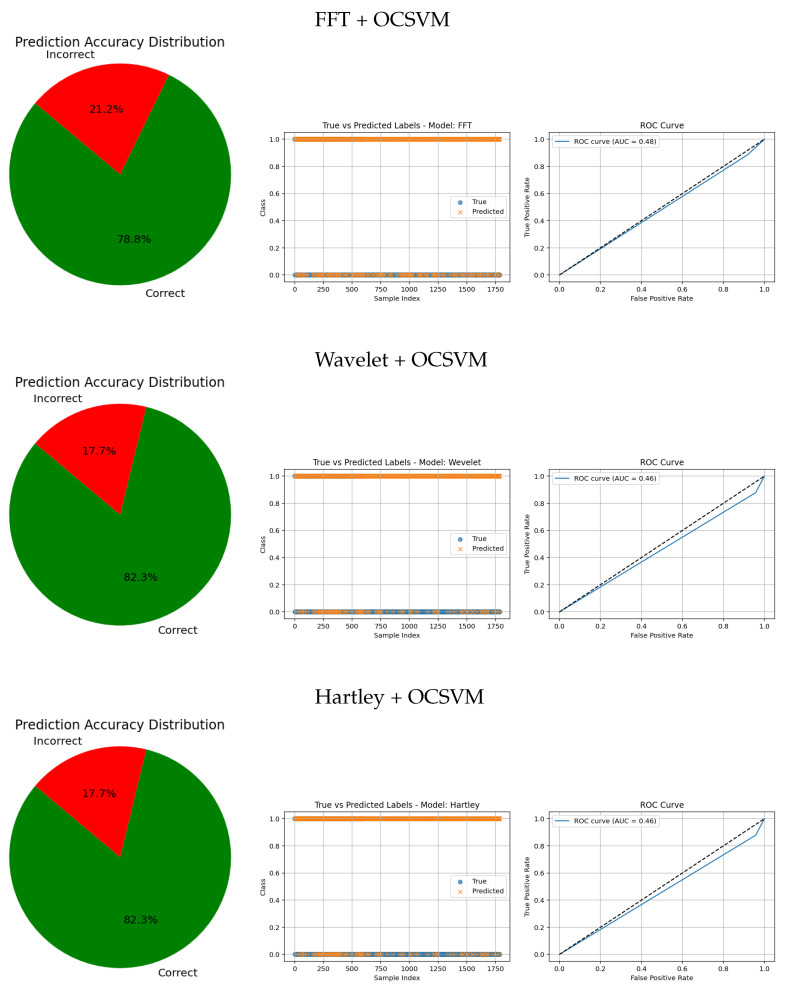
Comparative results of traditional feature-based methods combined with anomaly detection approaches. Each row shows the prediction accuracy distribution, predicted labels, and ROC curve.

**Table 1 jimaging-11-00207-t001:** Standardized summary of CNN architectures and feature extraction flow.

CNN Model	Main Layer Flow (up to GAP)	Vector After GAP	Additional Layers	Final Vector
**Custom CNN**	Conv2D(32) → MaxPool → Conv2D(64) → MaxPool → Conv2D(128) → GAP	(None, 128)	Dropout(0.3) → Dense(128, ReLU)	(None, 128)
**VGG16**	Convolutional blocks up to block5_conv3 → GAP	(None, 512)	Dropout(0.3) → Dense(128, ReLU)	(None, 128)
**MobileNetV2**	Depthwise separable blocks → Final bottleneck → GAP	(None, 1280)	Dropout(0.3) → Dense(128, ReLU)	(None, 128)
**InceptionV3**	Inception modules up to Mixed10 → GAP	(None, 2048)	Dropout(0.3) → Dense(128, ReLU)	(None, 128)
**DenseNet121**	Dense blocks + transition layers → GAP	(None, 1024)	Dropout(0.3) → Dense(128, ReLU)	(None, 128)

*Note:* Table 1 summarizes the architectural design of the custom CNN used as a feature extractor. Compared with standard pretrained models, such as VGG16 or InceptionV3, this lightweight architecture is tailored for the specific dataset and tasks of this study. Although it was also tested as a direct classifier, its primary role here is to generate representative feature vectors for the OCC framework.

**Table 2 jimaging-11-00207-t002:** Method and algorithm used, including input image dimensions and output size after algorithm application.

Method	Algorithm	Input Dimensions	Output Size
Input Image	OCSVM	150 × 150 × 3	67,500
	CNN	150 × 150 × 3	128
CNN + OCSVM	Custom CNN Backbone + OCSVM	150 × 150 × 3	128
	DenseNet121 Backbone + OCSVM	150 × 150 × 3	128
	VGG16 Backbone + OCSVM	150 × 150 × 3	128
	MobileNetV2 Backbone + OCSVM	150 × 150 × 3	128
	InceptionV3 Backbone + OCSVM	150 × 150 × 3	128
Dimensionality Reduction (DR)	Embedded + OCSVM	150 × 150 × 3	100
	PCA + OCSVM	150 × 150 × 3	100
FBM	Gradient Method + OCSVM	150 × 150 × 3	67,500
	Histogram Method + OCSVM	150 × 150 × 3	67,500
TFA	FFT + OCSVM	150 × 150 × 3	22,500
	Wavelet + OCSVM	150 × 150 × 3	90,000
	Hartley + OCSVM	150 × 150 × 3	22,500

**Table 3 jimaging-11-00207-t003:** Results of CNN and OCSVM using raw images as input: variation in parameter ν.

Algorithm	Pr (%)	Sens (%)	F1 (%)	Acc (%)	TP	FN	FP	TN
OCSVM (ν=0.01)								
Class 1 (Healthy)	60.27 (326/541)	78.53 (326/415)	68.13	63.04 (519/823)	326	89	215	193
Class 2 (Tumorous)	68.44 (193/282)	47.34 (193/408)	56.04	63.04 (519/823)	193	215	89	326
CNN								
Class 1 (Healthy)	98.02 (396/404)	98.26 (396/403)	98.14	97.83 (675/690)	396	7	8	279
Class 2 (Tumorous)	97.55 (279/286)	97.21 (279/287)	97.38	97.83 (675/690)	279	8	7	396

**Table 4 jimaging-11-00207-t004:** Performance metrics of different deep learning algorithms on classification of healthy and tumorous cases based on the standard confusion matrix.

Algorithm	Pr (%)	Sens (%)	F1 (%)	Acc (%)	TP	FN	FP	TN
InceptionV3 Backbone Network + OCSVM (ν=0.01):								
Class 1 (Healthy)	91.29 (304/333)	97.03 (304/313)	94.07	93.67 (562/600)	304	9	29	258
Class 2 (Tumorous)	96.63 (258/267)	89.97 (258/287)	93.18	93.67 (562/600)	258	29	9	304
DenseNet121 Backbone Network + OCSVM (ν=0.01):								
Class 1 (Healthy)	91.47 (311/340)	99.36 (311/313)	95.29	94.83 (569/600)	311	2	29	258
Class 2 (Tumorous)	99.23 (258/260)	89.97 (258/287)	94.37	94.83 (569/600)	258	29	2	311
MobileNetV2 Backbone Network + OCSVM (ν=0.01):								
Class 1 (Healthy)	88.57 (310/350)	99.04 (310/313)	93.53	92.83 (557/600)	310	3	40	247
Class 2 (Tumorous)	98.80 (247/250)	86.03 (247/287)	92.02	92.83 (557/600)	247	40	3	310
VGG16 Backbone Network + OCSVM (ν=0.1):								
Class 1 (Healthy)	92.54 (310/335)	99.04 (310/313)	95.68	95.33 (572/600)	310	3	25	262
Class 2 (Tumorous)	98.87 (262/265)	91.32 (262/287)	94.96	95.33 (572/600)	262	25	3	310
CNN Backbone Network + OCSVM (ν=0.01):								
Class 1 (Healthy)	77.36 (311/402)	99.36 (311/313)	87.37	84.50 (507/600)	311	2	91	196
Class 2 (Tumorous)	98.99 (196/198)	68.29 (196/287)	80.65	84.50 (507/600)	196	91	2	311

**Table 5 jimaging-11-00207-t005:** Evaluation metrics for dimensionality reduction methods combined with OCSVM.

Algorithm	Pr (%)	Sens (%)	F1 (%)	Acc (%)	TP	FN	FP	TN
Embedded + OCSVM (ν=0.1)								
Class 1 (Healthy)	99.18 (486/490)	32.40 (486/1500)	48.91	43.04 (769/1787)	486	1014	4	283
Class 2 (Tumorous)	21.82 (283/1297)	98.61 (283/287)	35.74	43.04 (769/1787)	283	4	1014	486
PCA + OCSVM (ν=0.01):								
Class 1 (Healthy)	81.17 (1117/1376)	74.46 (1117/1500)	77.66	64.11 (1145/1787)	1117	383	259	28
Class 2 (Tumorous)	6.82 (28/411)	9.76 (28/287)	8.01	64.11 (1145/1787)	28	259	383	1117

**Table 6 jimaging-11-00207-t006:** Evaluation metrics for histogram and gradient methods combined with OCSVM.

Algorithm	Pr (%)	Sens (%)	F1 (%)	Acc (%)	TP	FN	FP	TN
Gradient + OCSVM (ν=0.01)								
Class 1 (Healthy)	83.89 (198/236)	13.20 (198/1500)	23.00	25.00 (447/1787)	198	1302	38	249
Class 2 (Tumorous)	16.05 (249/1551)	86.75 (249/287)	27.18	25.00 (447/1787)	249	38	1302	198
Histogram + OCSVM (ν=0.01)								
Class 1 (Healthy)	83.81 (1401/1672)	93.40 (1401/1500)	88.36	79.44 (1430/1800)	1401	99	271	29
Class 2 (Tumorous)	22.66 (29/128)	9.67 (29/300)	13.47	79.44 (1430/1800)	29	271	99	1401

**Table 7 jimaging-11-00207-t007:** Evaluation metrics for FFT, Wavelet, and Hartley methods combined with OCSVM.

Algorithm	Pr (%)	Sens (%)	F1 (%)	Acc (%)	TP	FN	FP	TN
FFT + OCSVM (ν=0.01)								
Class 1 (Healthy)	90.51 (124/137)	8.27 (124/1500)	15.05	21.14 (378/1787)	124	1376	33	254
Class 2 (Tumorous)	15.58 (254/1630)	88.50 (254/287)	26.42	21.14 (378/1787)	254	33	1376	124
Wavelet + OCSVM (ν = 0.9)								
Class 1 (Healthy)	64.64 (64/99)	4.27 (64/1500)	7.91	17.68 (316/1787)	64	1436	35	252
Class 2 (Tumorous)	14.93 (252/1688)	87.80 (252/287)	25.46	17.68 (316/1787)	252	35	1436	64
Hartley + OCSVM (ν = 0.9)								
Class 1 (Healthy)	71.43% (60/84)	4.00% (60/1500)	7.54	18.07% (323/1787)	60	1440	24	263
Class 2 (Tumorous)	15.45% (263/1703)	91.63% (263/287)	26.27	18.07% (323/1787)	263	24	1440	60

**Table 8 jimaging-11-00207-t008:** Comparison of inference time (in milliseconds) and computational complexity among different models.

Model	Feature Extraction Time per Image (ms)	Prediction Time per Image (ms)	FLOPs per Image
InceptionV3 Backbone Network + OCSVM	16.43	0.00997	$2.52\times 10^{9}$
DenseNet121 Backbone Network + OCSVM	18.33	0.01164	$2.52\times 10^{9}$
MobileNetV2 Backbone Network + OCSVM	4.64	0.01164	$2.93\times 10^{8}$
VGG16 Backbone Network + OCSVM	91.09	0.00665	$1.32\times 10^{10}$
CNN Backbone Network + OCSVM	1.75	0.03989	$4.11\times 10^{8}$

**Table 9 jimaging-11-00207-t009:** Comparison of brain tumor classification methods across different studies.

Author(s)	Method/Description	Accuracy (%)
This work	CNN	97.83
This work	VGG16 Backbone Network with OCSVM (ν=0.1)	95.33
Cheng et al. [33]	Data augmentation, region partitioning, intensity histograms, GLCM, BoW	91.3
Ismael et al. [32]	Neural network using statistical descriptors	92.0
Tahir et al. [35]	2D Discrete Wavelet Transform (DWT) with Daubechies wavelet features	86.0
Paul et al. [38]	CNN-based model; performance improved with lower image resolution	90.3
Afshar et al. [34]	Capsule Network (CapsNet) with segmentation	86.7
Afshar et al. [34]	Capsule Network (CapsNet) without segmentation	78.0
Afshar et al. [39]	Refined CapsNet with fewer samples, robust to transformations	90.9
Zhou et al. [40]	Recurrent neural networks with automated region segmentation	92.1
Pashaei et al. [41]	CNN + Kernel Extreme Learning Machine (KELM) hybrid model	93.7
Abiwinanda et al. [36]	CNNs without segmentation (best config among 7)	84.2
Navid et al. [42]	GAN-based model with FC layers and data augmentation	95.6
Guo et al. [37]	CNN + graph-based model for PET scans (binary classification)	93.0
Guo et al. [37]	CNN + graph-based model (three-class classification)	77.0

## Data Availability

The used dataset is public in Kaggle [20].

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
