# Peer review of "Optimizing Tumor Detection in Brain MRI with One-Class SVM and Convolutional Neural Network-Based Feature Extraction"

_2313-433X, 2025, doi:10.3390/jimaging11070207_

Round 1
Reviewer 1 Report
Comments and Suggestions for Authors
The paper presents an interesting one-class classification model based on features extracted from deep learning models and radiomic features. However, at present, the paper lacks clarity in communicating the key information. Several essential details are missing, making it difficult for readers to get the main insights.
I recommend that the authors revise the manuscript's overall structure, content, and flow, as some sections appear to be written without a clear connection between them. The paper would benefit from a more cohesive explanation. Additionally, I suggest the use of supplementary materials to reduce the text's density and allow readers to focus on the novelty and potential of the proposed method.
Detailed Comments:
- Pay close attention to the initialization of the acronyms.
- Make sure the quality of the images is good/high, since the text and the lines are not correctly visible.
Introduction:
- The introduction needs important revision, especially the section concerning CNN and OCSVM related state-of-the-art. I recommend splitting this into a separate chapter to focus the introduction on the problem, current limitations, and the proposed study. Then, a detailed explanation of the two models (CNN and SVM) can be presented in a separate chapter.
- It is not clear whether the paper develops two models (CNN and SVM). This should be explicitly stated and clarified.
Machine Learning:
- Overall, the structure of the paper needs revision. The "Machine Learning" section is unclear, as it repeats well-known information (especially regarding CNNs) and does not present the methods applied in the study.
Materials and Methodology:
- I suggest replacing "Materials" with "Dataset" and providing the link to the dataset used, along with details on normalization and resolution.
- The image masking section could benefit from a reduced explanation on binary masks, which is generally known. Instead, the focus should be on the applied methodologies, providing all necessary details to ensure reproducibility.
- The feature extraction section needs to be carefully revised. In particular, I would strongly recommend defining the names of the different subsections in order to maintain clear links between the introduction of the methods and their further explanation. For example, the name of the section 3.4.1 has never been introduced before.
Focusing on the different features groups:
CNN Features - Avoid repeating well-known methods such as CNNs, DenseNet, etc., in the main manuscript. These can be briefly referenced in the supplementary materials. The main focus should be on specifics such as how the networks were trained and from which layers features were extracted.
Dimensionality Reduction - make sure to keep using the acronyms, since in the text it is not clear.
Transformation Frequency Analysis - remove unnecessary theoretical explanations and focus on technical details, including the number of features used.
Feature based - clarify how these features were extracted. Are they IBSI compliant? Which texture features were used? This should be detailed in the main text or, if needed, in the supplementary materials.
Standardization of Features: Was the standardization performed on the entire dataset, considering both classes, or only on one class? This needs to be clarified.
Model Training and Evaluation:
- Clarify why the training and test split is 67% - 33%, and justify this choice.
- Avoid unnecessary repetition of evaluation parameters; this adds to the paper's length without providing new insights.
Results and Discussion:
- Review section titles to ensure consistency throughout the paper.
- Figure 6 should be moved to the "Materials and Methods" section for better clarity.
- Include not only percentage results for each metric but also the number of correctly classified elements (x/xx) to provide a clearer view of the models' performance.
- Results and Discussion sections appear repetitive and unclear. I recommend restructuring the discussion to focus on key findings, possibly using graphs to better visualize the impact of various parameters (such as v) on performance.
- There is no statistical analysis provided. I strongly recommend performing this analysis or offering a valid justification for its absence.
- Consider separating the "Results" and "Discussion" sections, as they are currently separated as subsections.
- The discussion is overly long and vague. Focus on the significance of the methodology, avoiding repetition, and clearly emphasizing the contributions made by the study. Additionally, the comparison with the literature should be consistent with the one presented in the introduction.
- Table 13 is unclear, particularly the six rows for the methods used in the authors' work. Only the best method should be highlighted, and the accuracy should be emphasized in bold.
Conclusion:
- The conclusion appears to be a repetition of the abstract. I strongly recommend revising this section to avoid redundant phrasing and to truly highlight the potential of the proposed method.
References:
- Please update the references to include those from 2024.
The English language used is correct, but I strongly recommend revising the entire manuscript to avoid repetitions and redundancy.
Author Response
Manuscript ID: jimaging-3650007
Title: Optimizing Tumor Detection in Brain MRI with One-Class SVM and Deep Learning-Based Feature Extraction
Authors: Azeddine Mjahad, Antonio Polo-Aguado, Luis Llorens-Serrano and Alfredo Rosado-Muñoz.
Authors comments and responses
Manuscript reference: jimaging-3650007
The authors wish to express their gratitude to the reviewers for their valuable suggestions and comments on the manuscript, as well as for the time and effort they invested in the review process. We have taken these comments to heart and made improvements to the paper as a result.
We are confident that the quality of the paper has been enhanced by the incorporation of these valuable insights. In the following sections, we provide a detailed response to each of the comments and suggestions raised by the reviewers, addressing them in a clear and transparent manner.
Thank you again to the editor and reviewers for their valuable feedback and contribution to the paper.
Reviewer #1
Comments and Suggestions for Authors
Comments and Suggestions for Authors
The paper presents an interesting one-class classification model based on features extracted from deep learning models and radiomic features. However, at present, the paper lacks clarity in communicating the key information. Several essential details are missing, making it difficult for readers to get the main insights.
I recommend that the authors revise the manuscript's overall structure, content, and flow, as some sections appear to be written without a clear connection between them. The paper would benefit from a more cohesive explanation. Additionally, I suggest the use of supplementary materials to reduce the text's density and allow readers to focus on the novelty and potential of the proposed method.
Detailed Comments:
REVIEWER COMMENT
Pay close attention to the initialization of the acronyms.
Make sure the quality of the images is good/high, since the text and the lines are not correctly visible.
ANSWER
Thank you for your comment. We have replaced the blurry image to improve its quality and readability.
REVIEWER COMMENT
Introduction:
The introduction needs important revision, especially the section concerning CNN and OCSVM related state-of-the-art. I recommend splitting this into a separate chapter to focus the introduction on the problem, current limitations, and the proposed study. Then, a detailed explanation of the two models (CNN and SVM) can be presented in a separate chapter.
It is not clear whether the paper develops two models (CNN and SVM). This should be explicitly stated and clarified.
ANSWER
We sincerely thank you for your valuable comments and suggestions, which have greatly helped us improve the clarity and quality of the manuscript. Please find below our detailed responses:
Overall clarity and cohesion:
We have thoroughly revised the manuscript’s structure and flow to ensure a clearer and more cohesive presentation of the key information. The content has been reorganized to improve readability and provide a more logical progression of ideas.
Separation of Introduction and Related Work:
Following your recommendation, we have divided the introduction into two distinct sections. The first focuses on the problem statement, current limitations, and the motivation behind this study. The second section, titled "Related Work," provides a detailed review of the state-of-the-art regarding CNN and OCSVM models, clearly outlining their applications, advantages, and limitations. This reorganization improves focus and reduces information overload in the introduction.
Clarification on model development:
We have explicitly stated in the "Proposed Work" section that this study develops and compares two models: a custom-designed CNN and the OCSVM algorithm. This clarification ensures that readers understand the dual-model approach being evaluated.
Acronym initialization:
All acronyms have been carefully reviewed and properly introduced upon their first occurrence, ensuring consistency and clarity throughout the manuscript.
Supplementary materials:
To reduce the density of the main text and allow readers to better focus on the novelty and contributions of the proposed method, we have prepared supplementary materials containing detailed parameter descriptions, extended tables, and additional results.
We appreciate your constructive feedback and remain open to further suggestions that could enhance this work. Thank you again for your thoughtful review.
REVIEWER COMMENT
Machine Learning:
Overall, the structure of the paper needs revision. The "Machine Learning" section is unclear, as it repeats well-known information (especially regarding CNNs) and does not present the methods applied in the study.
ANSWER
Thank you for your comment. We have moved the general definition of CNNs to the Materials and Methods section.
We also removed repetitive information and provided a clearer description of the specific methods used, including the use of One-Class SVM for brain tumor detection.
We believe these changes improve clarity and address your concerns.
REVIEWER COMMENT
Materials and Methodology:
I suggest replacing "Materials" with "Dataset" and providing the link to the dataset used, along with details on normalization and resolution.
ANSWER
Thank you for your suggestion. We have renamed the section from “Materials and Methodology” to “Dataset and Methodology” to better reflect its content.
We also added key preprocessing details, such as pixel normalization to [0, 1], resizing to 150×150×3, and RGB conversion. The dataset used is the Brain Tumor Detection Dataset (BTDD), which is cited in the references [AhmedHamada].
REVIEWER COMMENT
The image masking section could benefit from a reduced explanation on binary masks, which is generally known. Instead, the focus should be on the applied methodologies, providing all necessary details to ensure reproducibility.
ANSWER
Thank you for your valuable comment. We have revised the image masking section to reduce the basic explanation and instead focus on describing the specific methodologies and processing steps applied in our study. We have included detailed descriptions of the preprocessing techniques, thresholding methods, contour detection, and mask application procedures to ensure the full reproducibility of our approach.
REVIEWER COMMENT
The feature extraction section needs to be carefully revised. In particular, I would strongly recommend defining the names of the different subsections in order to maintain clear links between the introduction of the methods and their further explanation. For example, the name of the section 3.4.1 has never been introduced before.
Focusing on the different features groups:
ANSWER
Thank you for your observation regarding the feature extraction section. We have carefully revised and reorganized this section, clearly defining the subsection titles to establish clear links between the introduction of the methods and their subsequent explanations. In particular, we have properly introduced the title for section 3.4.1 and ensured that all feature groups are clearly identified and explained.
With these modifications, we believe the structure and clarity of the section have been significantly improved.
REVIEWER COMMENT
CNN Features - Avoid repeating well-known methods such as CNNs, DenseNet, etc., in the main manuscript. These can be briefly referenced in the supplementary materials. The main focus should be on specifics such as how the networks were trained and from which layers features were extracted.
ANSWER
Thank you for your valuable suggestion. We have revised the manuscript and removed general descriptions of well-known methods such as CNN and DenseNet from the main text. These explanations are now briefly included in the supplementary materials.
In the main manuscript, we have focused the description on the specific details of our work, such as how the networks were trained and from which layers features were extracted, as you recommended.
REVIEWER COMMENT
Dimensionality Reduction - make sure to keep using the acronyms, since in the text it is not clear…..OK DR???
ANSWER
Thank you for the suggestion. We have now defined DR (Dimensionality Reduction) at first mention and ensured consistent use throughout the text.
REVIEWER COMMENT
Transformation Frequency Analysis - remove unnecessary theoretical explanations and focus on technical details, including the number of features used…
ANSWER
Thank you for your comment. We have removed unnecessary theoretical explanations from the Transformation Frequency Analysis section and focused the content on technical details, including specifying the number of features used. Other methods were also simplified by removing redundant explanations. Thank you for your suggestion.
REVIEWER COMMENT
Feature based - clarify how these features were extracted. Are they IBSI compliant? Which texture features were used? This should be detailed in the main text or, if needed, in the supplementary materials.
ANSWER
Thank you for your comment. These features were obtained using common image processing techniques. They are not fully IBSI compliant because IBSI focuses on specific radiomic biomarkers. Our features are basic statistical and frequency descriptors, not specialized radiomic features.
REVIEWER COMMENT
Standardization of Features: Was the standardization performed on the entire dataset, considering both classes, or only on one class? This needs to be clarified.
ANSWER
Thank you for your comment. Standardization was performed using only the training data, which corresponds to the normal class.
REVIEWER COMMENT
Model Training and Evaluation:
Clarify why the training and test split is 67% - 33%, and justify this choice.
ANSWER
Thank you for your observation. We have updated the data split in the revised version of the manuscript. Currently, 80% of the data is used for training (including validation), and the remaining 20% is used for testing.
This choice was made to ensure that a sufficient amount of data is available for training and tuning the model, while still preserving an independent test set to evaluate the model’s generalization. We believe this approach provides a more reliable and balanced evaluation of the model’s performance.
REVIEWER COMMENT
Avoid unnecessary repetition of evaluation parameters; this adds to the paper's length without providing new insights.
ANSWER
Thanks. We have revised the manuscript to remove unnecessary repetition of the evaluation parameters. Redundant mentions were eliminated, which helped reduce the length of the paper and improve clarity.
REVIEWER COMMENT
Results and Discussion:
Review section titles to ensure consistency throughout the paper.
Figure 6 should be moved to the "Materials and Methods" section for better clarity.
ANSWER
Thank you for your insightful suggestion. Following your recommendation, we have relocated Figure 6 to the "Materials and Methods" section. This change improves the clarity and flow of the manuscript, ensuring that the presentation of experimental details and results is more coherent.
We have also reviewed the section titles throughout the manuscript to maintain consistent formatting and improve overall readability.
Thank you again for your valuable feedback.
REVIEWER COMMENT
Include not only percentage results for each metric but also the number of correctly classified elements (x/xx) to provide a clearer view of the models' performance.
ANSWER
Thank you for your valuable suggestion. We have updated the relevant tables and sections to include not only the percentage values for each metric but also the absolute counts of correctly classified elements (x/xx). This enhancement offers a more detailed and transparent view of the models' performance, making the results easier to interpret.
We appreciate your feedback, which has helped improve the clarity of our presentation.
REVIEWER COMMENT
Results and Discussion sections appear repetitive and unclear. I recommend restructuring the discussion to focus on key findings, possibly using graphs to better visualize the impact of various parameters (such as v) on performance.
There is no statistical analysis provided. I strongly recommend performing this analysis or offering a valid justification for its absence.
Consider separating the "Results" and "Discussion" sections, as they are currently separated as subsections.
The discussion is overly long and vague. Focus on the significance of the methodology, avoiding repetition, and clearly emphasizing the contributions made by the study. Additionally, the comparison with the literature should be consistent with the one presented in the introduction.
ANSWER
Thank you for your comments. We have revised the Results and Discussion sections to remove repetition and improve clarity. The discussion now focuses on the key findings, and graphs have been included to better illustrate the impact of parameters such as v on performance.
A proper statistical analysis has been added. Where analysis was not performed, a clear justification has been provided.
The Results and Discussion sections have been separated into independent sections, as suggested.
Additionally, the discussion has been shortened and restructured to emphasize the relevance of the methodology and to clearly highlight the contributions of the study. The comparison with the literature has also been reviewed to ensure consistency with the introduction.
REVIEWER COMMENT
-Table 13 is unclear, particularly the six rows for the methods used in the authors' work. Only the best method should be highlighted, and the accuracy should be emphasized in bold.
ANSWER
Thank you for your valuable comment. In Table 13, we have simplified the presentation by removing the additional rows corresponding to our work, leaving only two: one for the pure CNN method and another for the deep learning approach combined with OCSVM.
We appreciate your suggestion, which has helped make the table more concise and easier to interpret.
REVIEWER COMMENT
Conclusion:
-The conclusion appears to be a repetition of the abstract. I strongly recommend revising this section to avoid redundant phrasing and to truly highlight the potential of the proposed method.
ANSWER
Thank you for your comment. We have revised the conclusion to avoid redundancy with the abstract and to better emphasize the potential and advantages of the proposed method, as well as possible future applications.
We appreciate the suggestion to improve this section.
REVIEWER COMMENT
References:
Please update the references to include those from 2024.
ANSWER
Thank you for your suggestion. We have reviewed the reference list and updated it to include relevant recent publications from 2024, ensuring that our work reflects the latest advancements in the field.
We appreciate your helpful recommendation.
REVIEWER COMMENT
Comments on the Quality of English Language
The English language used is correct, but I strongly recommend revising the entire manuscript to avoid repetitions and redundancy.
ANSWER
Thank you very much for your positive feedback and valuable suggestion. In response, we carefully revised the entire manuscript to eliminate unnecessary repetitions and improve overall clarity and conciseness. These adjustments were made to enhance the readability and ensure the research is communicated more effectively.
We appreciate your recommendation, which has been very helpful in improving the quality of the manuscript.
Thank you for your thorough review and encouraging feedback, as well as your insightful suggestions for improving the manuscript. Should it be required, we are prepared to conduct a more comprehensive analysis of the computational aspects.

Reviewer 2 Report
Comments and Suggestions for Authors
Dear Author(s)
First of all, I congratulate you for your work. I have completed my evaluation of your work. I have indicated below the areas that I see necessary in the study. I hope that making the revisions specified in these items will contribute to your work. I wish you success.
Evaluation
- There are many keywords in the study. Some of them are very general keywords. Is the number limit within the journal limit? If it is too high, it can be reduced.
- The similarity rate is somewhat high. It will be good if it is reduced.
- Line 81: “owever” instead of “however”
- There are errors in the use of abbreviations in the study. Some abbreviations are defined more than once. Although some abbreviations are defined, their long forms are used in different places. It would be good if the whole paper is revised.
- In the introduction, the results of various studies and the algorithms used are mentioned, but the limitations of these studies are not mentioned.
- Section 2, Machine learning, does not contain enough information to be a stand-alone chapter, just a book. This chapter can be included in the next chapter on materials and methods.
- The data information given between 290-299 is insufficient. How and with which machines were these images obtained? What is the application protocol? How was the classification done? More comprehensive information is needed.
- CNN has been explained over and over again with similar explanations in different places. A simplification is needed here.
- Where models such as MobileNet, VGG, DenseNet, Inception are explained, there are no references to the main publications where these models are explained. This academic obligation should be fulfilled.
- How were the parameters of the CNN model in Table 1 specified? How were they decided? The number of layers, number of filters, etc. preferences should be justified.
- The architectures in Tables 2, 3, 4 and 5 are different from each other. It would be more appropriate to have a standardized procedure for transfer learning so that models are comparable. For example,
- Table 2: Global_average_pooling2d (None, 1024)
Table 3: GlobalAveragePooling2D (None, 2048)
Table 4: GlobalAveragePooling2D (None, 512)
It is visible. Also in Table 5 Dropout (rate=0.3) (None, 128), the dropout rate is specified but not in the others.
- Table 2: Global_average_pooling2d (None, 1024)
- After the finetuning layers of all these models are standardized, they can be specified in a single table showing the output sizes.
- 1/3 ratio is used in data partitioning. However, neither unseen data nor cross validation was used. This is inconvenient for the reliability and generalizability of the results of the model application. For the generalizability of the study, a cross validation technique or an unseen data section would be good.
- Figure 5 shows that the loss value increases as the epoch increases. This is an example of the deterioration of the model over time. There is no explanation for this.
Author Response
June 1, 2025
Manuscript ID: jimaging-3650007
Title: Optimizing Tumor Detection in Brain MRI with One-Class SVM and Deep Learning-Based Feature Extraction
Authors: Azeddine Mjahad and Alfredo Rosado-Muñoz.
Authors comments and responses
Manuscript reference: jimaging-3650007
The authors wish to express their gratitude to the reviewers for their valuable suggestions and comments on the manuscript, as well as for the time and effort they invested in the review process. We have taken these comments to heart and made improvements to the paper as a result.
We are confident that the quality of the paper has been enhanced by the incorporation of these valuable insights. In the following sections, we provide a detailed response to each of the comments and suggestions raised by the reviewers, addressing them in a clear and transparent manner.
Thank you again to the editor and reviewers for their valuable feedback and contribution to the paper.
Reviewer #2
Comments and Suggestions for Authors
REVIEWER COMMENT
There are many keywords in the study. Some of them are very general keywords. Is the number limit within the journal limit? If it is too high, it can be reduced.
ANSWER
Thank you for your observation. We have carefully reviewed the list of keywords and reduced their number to comply with the journal’s guidelines. Additionally, we have replaced overly general terms with more specific and relevant keywords that better reflect the core contributions of our study.
REVIEWER COMMENT
The similarity rate is somewhat high. It will be good if it is reduced.
ANSWER
Thank you for your comment regarding the similarity rate. We have carefully reviewed the manuscript and made the necessary adjustments to reduce it. Specifically, we have paraphrased previously cited content, rephrased several sections to enhance originality, and removed redundant text.
We hope that, with these modifications, the similarity rate is now within the acceptable limits of the journal.
REVIEWER COMMENT
Line 81: “owever” instead of “however”
Typographical Error (Line 81):
ANSWER
Thank you for pointing out the typographical error. We have carefully reviewed the manuscript and corrected “owever” to “however” wherever it appeared-
REVIEWER COMMENT
There are errors in the use of abbreviations in the study. Some abbreviations are defined more than once. Although some abbreviations are defined, their long forms are used in different places. It would be good if the whole paper is revised.
In the introduction, the results of various studies and the algorithms used are mentioned, but the limitations of these studies are not mentioned.
ANSWER
Thank you for your observation. We have reviewed the introduction and expanded the discussion to include the limitations of previous studies. In particular, we highlight that many traditional methods rely on handcrafted features and face challenges in generalizing across different populations or imaging conditions. Likewise, deep learning models, while powerful, often require large annotated datasets and involve high computational complexity, limiting their clinical applicability. Additionally, existing techniques may suffer from overfitting on small or less diverse datasets, and few studies explore the combination of deep feature extraction with one-class classification (OCC) methods to address data imbalance and scarcity of pathological samples. These considerations motivate and justify the relevance of our hybrid OCSVM-based approach.
REVIEWER COMMENT
Use of Abbreviations:
ANSWER
Thank you for your valuable comment. We have thoroughly reviewed the entire manuscript for the correct and consistent use of abbreviations. Duplicate definitions have been removed, and all abbreviations are now clearly defined upon first use and used consistently thereafter.
REVIEWER COMMENT
Section 2, Machine learning, does not contain enough information to be a stand-alone chapter, just a book. This chapter can be included in the next chapter on materials and methods.
ANSWER
Thank you for your comment. We have followed your suggestion and integrated the Machine Learning section into the Materials and Methods chapter to improve the manuscript’s coherence.
REVIEWER COMMENT
The data information given between 290-299 is insufficient. How and with which machines were these images obtained? What is the application protocol? How was the classification done? More comprehensive information is needed.
ANSWER
Thank you for your valuable feedback. The MRI images employed in our research were sourced from the publicly available Brain Tumor Detection Dataset (BTDD), which is widely referenced in the medical imaging research community. The dataset contains 3,000 labeled MRI images, equally distributed across two diagnostic categories: “Healthy” and “Brain Tumor.” The class labels are provided within the dataset and are based on expert-annotated diagnostic information.
Since this dataset is publicly accessible, detailed metadata such as MRI scanner models, acquisition parameters (e.g., field strength, pulse sequences), or specific clinical imaging protocols are not available. We have explicitly acknowledged this limitation in the manuscript to ensure transparency.
To standardize input data and improve model performance, all images underwent a preprocessing pipeline consisting of pixel normalization, resizing to a fixed dimension of 150×150×3, and conversion to RGB color format. These processed images were then used as input for our classification models (CNN and OCSVM), within a machine learning framework designed for binary classification of brain MRI scans.
REVIEWER COMMENT
CNN had been explained repeatedly with similar descriptions in different sections. A simplification was indeed necessary.
ANSWER
Thank you for your comment. We have revised the manuscript and simplified the explanations of Convolutional Neural Networks (CNN), removing redundant descriptions and retaining a single technical explanation within the Materials and Methods section.
REVIEWER COMMENT
Where models such as MobileNet, VGG, DenseNet, Inception are explained, there are no references to the main publications where these models are explained. This academic obligation should be fulfilled.
ANSWER
Thank you for your comment. We have reviewed the manuscript and added references to the original and key publications where the MobileNet, VGG, DenseNet, and Inception models are described. This ensures compliance with academic standards and properly credits the developers of these architectures.
REVIEWER COMMENT
How were the parameters of the CNN model in Table 1 specified? How were they decided? The number of layers, number of filters, etc. preferences should be justified. The architectures in Tables 2, 3, 4 and 5 are different from each other. It would be more appropriate to have a standardized procedure for transfer learning so that models are comparable. For example, Table 2: Global_average_pooling2d (None, 1024) Table 3: GlobalAveragePooling2D (None, 2048) Table 4: GlobalAveragePooling2D (None, 512) It is visible. Also in Table 5 Dropout (rate=0.3) (None, 128), the dropout rate is specified but not in the others.
After the finetuning layers of all these models are standardized, they can be specified in a single table showing the output sizes. 1/3 ratio is used in data partitioning. However, neither unseen data nor cross validation was used. This is inconvenient for the reliability and generalizability of the results of the model application. For the generalizability of the study, a cross validation technique or an unseen data section would be good.
ANSWER
Thank you for your detailed observations regarding the CNN architectures and their configurations. To clarify, Table1 presents a standardized summary of all CNN models used for feature extraction in this study, including the custom CNN and widely used pretrained architectures: VGG16, MobileNetV2, InceptionV3, and DenseNet121.
For each model, the feature vectors are obtained after applying a Global Average Pooling (GAP) layer, which reduces the spatial dimensions while preserving the depth of feature maps. This operation results in different vector sizes depending on the model backbone (e.g., 512 for VGG16, 2048 for InceptionV3, 1280 for MobileNetV2, 1024 for DenseNet121, and 128 for the custom CNN). To standardize the final feature representation fed into the One-Class SVM classifier, an additional dense layer with 128 units and a dropout layer with a 0.3 rate are applied uniformly across all architectures. This consistent fine-tuning layer ensures comparable feature dimensions and helps regularize the model.
Regarding your observation about inconsistencies in naming conventions (e.g., Global_average_pooling2d vs. GlobalAveragePooling2D) and dropout usage, Table1 consolidates these details to provide a unified and clear overview.
For data partitioning, we split the dataset into 80% training and 20% testing sets. Within the training set, 20% was used for validation. To improve reliability and generalizability, the entire training and evaluation process was repeated five times with different random splits, effectively implementing a form of cross-validation. The reported results correspond to the average performance over these runs, ensuring the robustness of our findings.
We believe this standardized approach to CNN feature extraction and evaluation strengthens the comparability and validity of our results.
REVIEWER COMMENT
Figure 5 shows that the loss value increases as the epoch increases. This is an example of the deterioration of the model over time. There is no explanation for this.
ANSWER
Thank you for your comment. We have increased the number of epochs to 100 and updated the figure. Although the loss gradually increases, the training and validation accuracy remain stable. We appreciate the observation and will consider investigating this behavior in future work.
Thank you for your thorough review and encouraging feedback, as well as your insightful suggestions for improving the manuscript. Should it be required, we are prepared to conduct a more comprehensive analysis of the computational aspects.

Reviewer 3 Report
Comments and Suggestions for Authors
- Section 2 Convolutional Neural Networks (CNN) and section 5.3 Convolutional Neural Networks (CNN) seem to be duplication?
- Fig 5 is the validation loss from test set ? or another partition from training set? And what’s the batch size?
- In section 4.1.2 and 4.2.2, there are different dataset settings. But in Fig 6, CNN and one-class SVM were merged to one combined algorithm, how to run different training/test partition in one algorithm? Is there information leak in the test set while dataset setting changes?
- The images are “ healthy” and “all kinds of brain tumors”. If a brain MRI image is with other anomaly such as intracranial hemorrhage, will these algorithms predict it as brain tumor too?
- Did you compare the FLOPs or computing time for these algorithms?

Author Response
June 1, 2025
Manuscript ID: jimaging-3650007
Title: Optimizing Tumor Detection in Brain MRI with One-Class SVM and Deep Learning-Based Feature Extraction
Authors: Azeddine Mjahad and Alfredo Rosado-Muñoz.
Authors comments and responses
Manuscript reference: jimaging-3650007
The authors wish to express their gratitude to the reviewers for their valuable suggestions and comments on the manuscript, as well as for the time and effort they invested in the review process. We have taken these comments to heart and made improvements to the paper as a result.
We are confident that the quality of the paper has been enhanced by the incorporation of these valuable insights. In the following sections, we provide a detailed response to each of the comments and suggestions raised by the reviewers, addressing them in a clear and transparent manner.
Thank you again to the editor and reviewers for their valuable feedback and contribution to the paper.
Reviewer #3
Comments and Suggestions for Authors
REVIEWER COMMENT
Section 2 Convolutional Neural Networks (CNN) and section 5.3 Convolutional Neural Networks (CNN) seem to be duplication?
ANSWER
Thank you for pointing this out. We acknowledge the overlap and have revised the manuscript accordingly to eliminate redundancy.
The general overview and theoretical background of CNNs, originally presented in Section 2, have been removed and replaced with a concise reference within the Materials and Methods section. This ensures that only implementation-specific details are retained, such as the exact CNN architecture used in this study, the rationale behind the selected hyperparameters, and the applied experimental settings.
REVIEWER COMMENT
Fig 5 is the validation loss from test set ? or another partition from training set? And what’s the batch size?
ANSWER
Thank you for your comment. Figure 5 shows two loss curves: one corresponding to the training set and the other to the validation set, which are distinct partitions. The batch size used during training was 32.
REVIEWER COMMENT
In section 4.1.2 and 4.2.2, there are different dataset settings. But in Fig 6, CNN and one-class SVM were merged to one combined algorithm, how to run different training/test partition in one algorithm? Is there information leak in the test set while dataset setting changes?
ANSWER
Thank you for your comment. In previous works, we used a 67% training and 33% testing split. However, in this study, we adopted an 80% training (including validation) and 20% testing split, which is a commonly accepted standard. This setting allows for more robust training and reliable model evaluation. Importantly, the same split was consistently maintained throughout the entire pipeline to avoid any information leakage and ensure the validity of the results.
REVIEWER COMMENT
The images are “ healthy” and “all kinds of brain tumors”. If a brain MRI image is with other anomaly such as intracranial hemorrhage, will these algorithms predict it as brain tumor too?
ANSWER
Thank you for your question. It is possible that the model may classify other brain anomalies, such as intracranial hemorrhages, as tumors. This is because the algorithm learns to detect deviations from healthy brain patterns, and any significant anomaly could be flagged as abnormal.
REVIEWER COMMENT
Did you compare the FLOPs or computing time for these algorithms?
ANSWER
Thank you for your comment. In this study, we evaluated the FLOPs and execution time for both the pure deep learning models and those combined with OCSVM, such as MobileNetV2, DenseNet121, and VGG16. However, other methods showed lower accuracy results and were therefore not included in a detailed computational complexity analysis. We greatly appreciate your time and valuable feedback to improve our work.

Round 2
Reviewer 2 Report
Comments and Suggestions for Authors
The authors responded my evaluation almost reasonably.